# Tumor-Derived Antigenic Peptides as Potential Cancer Vaccines

**DOI:** 10.3390/ijms25094934

**Published:** 2024-04-30

**Authors:** Stanislav Sotirov, Ivan Dimitrov

**Affiliations:** Drug Design and Bioinformatics Lab, Faculty of Pharmacy, Medical University of Sofia, 2, Dunav Str., 1000 Sofia, Bulgaria; 113660@students.mu-sofia.bg

**Keywords:** cancer, antigen, immunogenicity, vaccine, bioinformatics

## Abstract

Peptide antigens derived from tumors have been observed to elicit protective immune responses, categorized as either tumor-associated antigens (TAAs) or tumor-specific antigens (TSAs). Subunit cancer vaccines incorporating these antigens have shown promise in inducing protective immune responses, leading to cancer prevention or eradication. Over recent years, peptide-based cancer vaccines have gained popularity as a treatment modality and are often combined with other forms of cancer therapy. Several clinical trials have explored the safety and efficacy of peptide-based cancer vaccines, with promising outcomes. Advancements in techniques such as whole-exome sequencing, next-generation sequencing, and in silico methods have facilitated the identification of antigens, making it increasingly feasible. Furthermore, the development of novel delivery methods and a deeper understanding of tumor immune evasion mechanisms have heightened the interest in these vaccines among researchers. This article provides an overview of novel insights regarding advancements in the field of peptide-based vaccines as a promising therapeutic avenue for cancer treatment. It summarizes existing computational methods for tumor neoantigen prediction, ongoing clinical trials involving peptide-based cancer vaccines, and recent studies on human vaccination experiments.

## 1. Introduction

Cancer vaccines work by instructing the immune system to recognize tumor antigens as foreign [1]. They can be used prophylactically to stop or prevent tumor development or therapeutically to treat patients who have already been diagnosed with cancer [2]. FDA-approved cancer vaccines are presented in Table 1.

Antigen selection strategy is the most import stage in the cancer vaccine development process. The ideal antigen should only be expressed by cancer cells and be highly immunogenic. The chosen antigen should also be present on all cancer cells which play a crucial role in cancer cell survival and protect against immune escape by mutations or loss of antigens in tumor cells. In order to induce an immune response, an antigen needs to be processed, presented to, and recognized by the immune cells [3]. Antigen processing and presentation refer to the processes that occur within a cell and result in fragmentation (proteolysis) of proteins, association of the fragments with MHC (major histocompatibility complex) molecules, and expression of the peptide–MHC molecules (p-MHC) at the cell surface where they can be recognized by the TCR (T-cell receptor) on a T cell (Figure 1) [4]. The TCR can recognize an antigen only in the form of a peptide bound to an MHC molecule on a human cell’s surface. The antigens recognized by T cells are peptides that arise from the breakdown of macromolecular structures, the unfolding of individual proteins, and their cleavage into short fragments through antigen processing. The peptides must be bound by an MHC molecule and presented at the cell surface. There are two classes of MHC molecules, MHC-I (MHC-Class I) and MHC-II (MHC-Class II). MHC-I molecules are specialized for the presentation of peptides derived from endogenous proteins (intracellular antigens) to the TCR of CD8+ T cells (CD8-expressing T cells) whilst MHC-II molecules are specialized for the presentation of extracellular antigens to the TCR of CD4+ T cells (CD4-expressing T cells) [5,6,7]. All T-cell epitopes can bind to MHC molecules; however, not all MHC binders are T-cell epitopes [8]. Feltkamp et al. showed that the binding affinity to MHC class I molecules is required but does not ensure T-cell immune responses [9]. Furthermore, factors other than MHC binding affinity are found to strongly influence T-cell immune responses, compared with the only moderate influence of MHC binding affinity [10]. Non-immunogenic epitopes may result from the following reasons: (a) p-MHC is truly unrecognized by TCR, (b) peptides are not presented by MHC, and (c) negative selection/clonal presentation is induced by excessive similarity to autologous peptides [11,12]. It has been estimated that only 1 in 200 potential peptide antigenic determinants will bind to a given MHC class I molecule with sufficient strength to elicit an immune response [13].

Several mechanisms alter the immune response against cancer cells, one of which is antigen spreading [14]. This phenomenon refers to the expansion of the immune responses from initially targeting one specific antigen (epitope) associated with the tumor to recognizing and attacking other antigens expressed by the same tumor or associated with it. As the immune system targets and eliminates tumor cells expressing the primary antigen, other tumor cells with different antigens may survive. As the tumor evolves, the immune system may begin to recognize these newly emerged antigens, leading to a secondary immune response. This secondary response involves the activation and expansion of immune cells specific to these new antigens. Epitope spreading can lead to a more robust and effective immune response against the tumor. By targeting multiple antigens, the immune system can overcome tumor heterogeneity and eliminate a broader range of cancer cells. The endogenous response of each individual patient against tumor antigens is considered a key factor in many current anticancer treatments [15]. These responses involve the recognition and elimination of cancerous or transformed cells by the immune system, particularly through the actions of cytotoxic T lymphocytes (CTLs) and natural killer (NK) cells. However, tumor cells employ defense mechanisms against the immune response, including restricting antigen recognition, immune system inhibition, and inducing T-cell exhaustion [16]. Hence, understanding the mechanisms underlying these responses is essential for developing immunotherapeutic strategies to enhance anti-tumor immunity and improve outcomes for cancer patients.

## 2. Types of Tumor Antigens

Peptide cancer vaccines are based on the epitope peptides that can elicit humoral and cellular immune responses targeting tumor-associated antigens (TAAs) or tumor-specific antigens (TSAs) [17]. The main characteristics of TAAs and TSAs are presented in Table 2.

TAAs are derived from self-antigens, meaning they are present in both cancerous and non-cancerous cells, but they may be expressed at higher levels or have altered post-translational modifications in cancer cells. As a result, the immune system can recognize these antigens as potential targets for immune responses specifically directed against cancer cells. TAAs play a crucial role in the development of cancer vaccines. Peptides derived from TAAs can be used as antigens in cancer vaccines to stimulate an immune response specifically targeting tumor cells. By presenting TAA-derived peptides to immune cells, such as T cells, the vaccines aim to activate and mobilize the immune system to selectively attack and eliminate cancer cells bearing these antigens. A major disadvantage of TAAs is that since they are self-antigens in their nature, a potential break of immune self-tolerance may occur, thus leading to autoimmune reactions. There are three main groups of TAAs. Overexpressed antigens are a large and diverse group that includes any protein found at increased levels in tumors compared with normal healthy cells and tissues [18]. One such widely studied antigen is Wilms tumor 1 (WT1). It is a well-known TAA expressed in various types of cancers, including acute myeloid leukemia (AML). It has been targeted in T-cell receptor (TCR) gene therapy and shown to prevent AML relapse post-transplant [19]. Another example is MART-1 (Melan-A). That antigen is a TAA predominantly expressed in melanoma. It has been studied extensively as a target for T-cell-based immunotherapies in melanoma patients [20]. The second group of TAAs consists of differentiation antigens [20]. They are selectively expressed by the cell lineage from which malignant cells evolved. One representative example, the prostate-specific antigen, has highly restricted distribution and is expressed in normal epithelial cells of the prostate gland, as well as prostate carcinomas [21]. These two kinds of tumor antigens are currently proved not to fit well in the landscape of cancer vaccines, with high immunological tolerance and toxicity that threaten the efficacy and safety of cancer vaccines for patients [20]. In regard to this, the third group, cancer germline antigens (CGAs, also known as cancer/testis antigens (CTAs)), stand aside among other TAAs. Besides their nonspecific tumor expression, they have only been found to be expressed in immune-privileged tissues. Thus, their aberrant expression in tumors makes them highly immunogenic. In addition to segregation by tissue barriers, trophoblastic and male germ cells, which are a normal localization of CTAs, lack HLA class I molecules expression and therefore cannot present antigens to T cells [22,23,24]. NY-ESO-1 is a well-known CTA expressed in various tumor types, including melanoma, lung cancer, and ovarian cancer. It has been investigated as a potential target for cancer vaccines and adoptive T-cell therapies [25]. MAGE-A3 (melanoma antigen A3) is another CTA that has been extensively studied as a target for immunotherapeutic approaches in multiple cancer types, including non-small cell lung cancer and melanoma [26].

Tumor-specific antigens (TSAs, also known as neoantigens) are a subset of antigens that are uniquely expressed by cancer cells but not found in normal cells. These antigens result from tumor-specific genetic alterations, such as somatic mutations, chromosomal rearrangements, or viral oncogene expression. They are highly specific to each individual. TSAs play a crucial role in cancer immunotherapy, particularly in the development of targeted cancer vaccines. Due to their tumor-specific expression, TSAs are attractive targets for therapeutic interventions aimed at stimulating an immune response against cancer cells while sparing healthy tissues, and due to their “non-self” features, they have a higher affinity to MHC molecules and T-cell receptors [27]. The specificity and selectivity of TSAs as targets for therapeutic interventions are their main advantages over TAAs. The field of neoantigen research is rapidly evolving, and new discoveries are continually being made. One of the most comprehensively studied neoantigens is KRAS. KRAS is a frequently mutated gene in various cancers, including pancreatic cancer, colorectal cancer, and lung adenocarcinoma. The G12V mutation in KRAS results in a constitutively active protein that drives oncogenic signaling pathways. This mutation generates a neoantigen that has been targeted in preclinical and clinical studies using personalized cancer vaccines [28] and adoptive T-cell therapies [29]. BRAF is another commonly mutated gene in cancers such as melanoma and colorectal cancer. The V600E mutation in BRAF leads to constitutive activation of the MAPK signaling pathway. This mutation generates a neoantigen that has been targeted using immune checkpoint inhibitors [30], adoptive T-cell therapies [31], and peptide-based vaccines [32], with promising results in clinical trials. EGFRvIII is a mutant variant of the epidermal growth factor receptor (EGFR) that is frequently observed in glioblastoma, a type of brain cancer. EGFRvIII arises from an in-frame deletion of exons 2-7, resulting in a truncated, constitutively active protein. This alteration creates a neoantigen that has been targeted in clinical trials using peptide-based vaccines [33] and adoptive T-cell therapies [34].

Also included in the group of TSAs are oncogenic viral antigens. In some cases, tumor cells may express antigens derived from viral proteins that are only present in infected cells. These antigens are attractive targets for immunotherapy because they are selectively expressed by tumor cells and can induce immune responses against both the virus and the tumor. Examples include the viral oncoproteins E6 and E7 derived from high-risk human papillomavirus (HPV) strains in HPV-associated cancers, such as cervical cancer [35].

## 3. Neoantigen Discovery and Selection for Peptide-Based Cancer Vaccines

Vaccination with TAAs has encountered limited efficacy, primarily attributed to their low tumor specificity, as these antigens are not exclusive to tumor cells [36]. Moreover, their utilization as cancer treatment candidates is hindered by suboptimal immune responses and the potential risk of autoimmune reactions. In contrast, TSAs, being exclusively expressed in tumor cells, offer a promising alternative. TSAs enable the development of personalized vaccines, providing an opportunity for selective tumor eradication while minimizing damage to healthy tissue.

The process of neoantigen discovery and selection for cancer treatment involves several steps to identify and prioritize potential neoantigens that can be targeted for immunotherapy [37] (Figure 2). The first step is to obtain tumor tissue or tumor-infiltrating lymphocytes (TILs) from the patient. The tumor DNA or RNA is then sequenced using techniques like whole-exome sequencing (WES) [38] or RNA sequencing (RNA-seq) [39] to identify genetic alterations and transcriptomic changes in the tumor cells. The sequencing data are analyzed to identify somatic mutations, including single-nucleotide variants (SNVs), insertions/deletions, and gene fusions, which result in the generation of potential neoantigens. These mutations are compared to the patient’s germline DNA to filter out non-tumor-specific alterations. In silico algorithms are used to predict and prioritize neoantigen candidates based on several factors [40]. These include binding affinity to human leukocyte antigen (HLA) molecules, processing and presentation capabilities, and potential immunogenicity.

Tumors typically harbor an average of over 100,000 mutations, emphasizing the vast number of potential neoantigens that may arise [41]. However, evaluating all these neoantigens experimentally is neither feasible nor practical. In response, in silico prediction algorithms play a crucial role by significantly reducing the pool of neoantigens for experimental validation. These algorithms utilize various information sources, such as amino acid sequences binding to the HLA, sequences recognized by the TCR, or the significance of specific amino acids in stabilizing the HLA–peptide–TCR complex [42]. They rely on principles of sequence alignment [43], machine learning [44], and artificial intelligence [45] to predict neoantigens accurately and efficiently. The latest advancements in neoantigen prediction primarily focus on supervised learning techniques, leveraging known data of both neoantigens and non-neoantigens. These approaches involve building predictive models capable of distinguishing between these two classes. Through supervised learning, these models are trained on labeled datasets and subsequently evaluated using validation datasets to measure their predictive accuracy, sensitivity, specificity, and other performance metrics [46]. Once trained and validated, these models are then deployed to predict candidate neoantigens efficiently and reliably. 

Various computational tools (Table 3) are commonly used for neoantigen prediction. HLA typing of the patient is performed to determine the specific HLA alleles expressed by their immune cells [47]. The process is performed by in silico prediction tools [48]. This information is crucial for predicting neoantigen presentation and the subsequent immune response. The predicted neoantigens are further validated through experimental methods, such as peptide–HLA binding assays [49] or mass spectrometry [50], to confirm their binding affinity to HLA molecules and their presentation on the tumor cell surface. These assays help refine the list of potential neoantigens and prioritize those with high immunogenicity and HLA binding. Selected neoantigens can be further evaluated through functional assays to assess their ability to elicit immune responses. This may involve in vitro studies [51] using T cells from the patient or healthy donors, assessing T-cell reactivity against the neoantigen-expressing tumor cells. In vivo studies [52] using animal models or patient-derived xenograft (PDX) models can also provide insights into neoantigen-specific immune responses and their impact on tumor growth. Finally, the most promising neoantigens are considered for clinical translation, such as the development of personalized cancer vaccines or adoptive cell therapies. These therapies involve the administration of neoantigen-specific vaccines [53] or the genetic modification of patient-derived T cells to express neoantigen-specific TCRs [54] or chimeric antigen receptors (CARs) [55]. 

Personalized peptide-based cancer vaccines are still in the early stages of development and are primarily being tested in clinical trials (Table 4). Numerous challenges persist in the advancement and extensive application of this type of immunotherapy [68]. The protracted process encompassing cancer genome sequencing, neoantigen detection, and validation of immunogenicity contributes to the substantial financial burden associated with personalized vaccines, impeding their feasibility for broad implementation in cancer patients. The emergence of enhanced bioinformatics tools tailored for neoantigen characterization, deeper insights into tumor immunology, and advancements in vaccine formulation and administration techniques are imperative for the realization and utilization of innovative neoantigen-based cancer vaccines.

## 4. Studies with Tumor Antigens

To date, there are few confirmed neoantigens in humans. While there are abundant data from lab tests and animal studies, applying these findings to human cancer patients is complex due to differences in our immune systems. Yet, recent human vaccination experiments bring hope that soon there will be more comprehensive studies. The studies on cancer epitopes that have been validated in vivo, along with other studies providing valuable insights into this field of research, are presented in Table 5. 

Most studies focus on finding patient-specific neoantigens and checking the immune response against them in real patients. Melanoma is a primary focus due to its high mutation rate [91], followed by lung cancer [92]. However, pancreatic cancer patients with longer survival times tend to have stronger T-cell activity and fewer immunogenic mutations, highlighting the importance of neoantigen quality in therapy [93]. While these three types of cancer are the primary focus of human immunization experiments, other cancers investigated in these studies typically have lower tumor mutational burdens, and neoantigen vaccinations against them are mostly reported as single-case studies.

### 4.1. Melanoma Studies

Khong and colleagues [69] investigated a vaccination approach in 37 patients with metastatic melanoma. They used epitopes from the NY-ESO-1 antigen presented by HLA-A0201 and/or HLA-DPbeta104 molecules. Interestingly, the vaccination not only targeted NY-ESO-1 but also triggered responses against other melanoma antigens, suggesting a broader anti-tumor immune reaction. This phenomenon, known as antigen spreading, was further explored by Corbiere et al. [70], who studied the mechanisms behind melanoma metastasis regression post-vaccination. They found that antigen-specific T cells could recognize various tumor antigens beyond the original targets.

Carreno et al. [71] demonstrated an increase in natural neoantigen-specific immunity and identified new HLA class I-restricted neoantigens in advanced melanoma patients after receiving a dendritic cell vaccine loaded with neoantigens. They confirmed the presentation of these neoantigens in human melanoma using mass spectrometry, revealing previously unrecognized targets.

Cohen et al. [72] developed a method to isolate neoantigen-specific T cells from both tumor and peripheral blood. They identified nine mutated epitopes from five out of eight patients using whole-exome sequencing and peptide–MHC tetramers. They successfully isolated mutation-reactive T cells from peripheral blood, showing reactivity towards eight out of the nine identified epitopes.

Linette et al. [73] explored how intratumoral heterogeneity (ITH) influences T-cell immunity in melanoma. They found clonal and subclonal neoantigens in multiple metastases from four melanoma patients. Although CD8+ T-cell responses against these neoantigens were limited to a subset with a restricted TCR repertoire, mature dendritic cell vaccination using tumor-encoded amino acid-substituted (AAS) peptides revealed diverse CD8+ T-cell responses specific to neoantigens. These findings highlight the importance of therapeutic vaccination because many T-cell clones that can recognize melanoma neoantigens are in a naive state and may not be easily identified.

Hu et al. [74] studied personalized neoantigen vaccines in eight patients with surgically resected stage IIIB/C or IVM1a/b melanoma. They observed tumor infiltration by neoantigen-specific T-cell clones post-vaccination, along with epitope spreading.

Sahin et al. [75] explored personalized RNA-based vaccines targeting distinct tumor mutations in 13 patients. By administering tumor-specific neoantigens, they aimed to induce a comprehensive immune response against cancer cells. These personalized RNA mutanome vaccines hold promise for revolutionizing cancer treatments through their tailored approach.

### 4.2. Pancreatic Carcinoma Studies

The study by Abou-Alfa et al. [76] aimed to test a vaccine targeting KRAS codon 12 mutations, often seen in pancreatic adenocarcinoma, as potential tumor-specific neoantigens. They vaccinated 24 patients who had their pancreatic cancer removed and showed KRAS mutations with a 21-mer peptide vaccine containing their tumor’s KRAS mutation. Results showed the vaccine was safe, with no severe side effects, but it did not trigger strong immune responses against the KRAS mutation. Median recurrence-free survival was 8.6 months, and median overall survival was 20.3 months, indicating limited effectiveness. 

Dillard et al. [77] explored TCR redirected T cells as a therapy for solid tumors, focusing on KRAS mutations. They identified and studied four specific KRAS-targeting T cells from a pancreatic cancer patient who had received a vaccine with mutated KRAS peptides. These T cells were able to recognize and attach to cells presenting KRAS peptides, showing promising potential. Some T cells could recognize multiple KRAS mutations, suggesting they could be effective against different tumor types. The study also confirmed that processed KRAS peptides were effectively presented to T cells, supporting the idea that KRAS mutations could be good targets for cancer treatment.

Sonntag et al. [78] reported on a patient with metastasized pancreatic ductal carcinoma who received personalized neoantigen-derived multipeptide vaccines as treatment. These vaccines targeted the patient’s unique neoantigens and were given as therapy. The outcome showed that the four-peptide vaccine triggered a strong and lasting immune response against the neoantigens. Importantly, this immune response was associated with a prolonged period of clinical remission for the patient.

Chen et al. [79] conducted a study to investigate personalized neoantigen-based immunotherapy for advanced pancreatic cancer patients who did not respond to standard treatments. They designed personalized cancer vaccines from neoantigens, with each patient receiving up to 20 neoantigen peptides. These vaccines, called iNeo-Vac-P01, were given to seven patients with low tumor mutation burden. The study found that iNeo-Vac-P01 was safe, with no severe side effects. Patients treated with iNeo-Vac-P01 had a mean overall survival of 24.1 months compared to 8.3 months without the vaccine. Progression-free survival averaged 3.1 months. Notably, one patient who survived for 21 months after vaccination showed a significant increase in antigen-specific TCR clones afterward.

### 4.3. Glioblastoma Studies

Keskin et al. [80] investigated using personalized neoantigen vaccines to treat glioblastoma, a type of brain tumor. They found that patients who did not receive dexamethasone, a powerful corticosteroid often used for brain swelling, had immune responses both in the body and within the tumor. These responses involved CD4+ and CD8+ T cells targeting neoantigens. However, despite these immune reactions, all ten patients in the study had their tumors come back and eventually passed away due to the disease getting worse. This suggests that while the immune responses were triggered, they faced significant hurdles in effectively fighting the tumors.

Johanns et al. [81] conducted a study primarily to assess the effectiveness of a personalized vaccine treatment plan. This plan involved giving the patient an autologous tumor lysate-dendritic cell vaccine (DCVax-L) first, followed by a neoantigen-based synthetic long peptide vaccine (GBM.PVax). The patient they studied was a 66-year-old woman diagnosed with GBM. After receiving the vaccines, they examined the patient’s blood and tumor-infiltrating lymphocytes (TILs). They found clear responses from both CD8+ and CD4+ T cells specifically targeting the neoantigens induced by the peptide vaccine. By analyzing the tumor’s genetic and transcriptomic characteristics before and after treatment, they discovered evidence of how the tumor evolved and possible ways it evades the immune system. This highlights the complexity of interactions between tumors and the immune system in GBM.

### 4.4. Lung Cancer Studies

Li et al. [82] presented a detailed case report of an Asian lung cancer patient who received personalized peptide vaccination (PPV) targeting specific neo-epitopes. After starting PPV, the patient showed significant clinical improvement with rapid tumor shrinkage. Impressively, within just three to four months, there was a substantial reduction in tumor size and positive clinical outcomes. Importantly, the personalized neo-epitope vaccination was well tolerated, with no significant treatment-related side effects. These findings highlight the potential effectiveness and safety of personalized neo-epitope vaccination as a treatment option for lung cancer patients. Further research and clinical trials are needed to confirm these results.

Li et al. [83] conducted a phase I clinical trial to evaluate the feasibility, safety, and effectiveness of personalized Neoantigen (NeoAg) peptide vaccination (PPV) in patients with stage III/IV non-small cell lung cancer (NSCLC) and EGFR mutations. Out of 29 enrolled patients, 24 received the vaccine. The median progression-free survival was 6.0 months, and the overall survival was 8.9 months. Notably, within 3-4 months of starting PPV, seven patients showed objective clinical responses, including one complete response. All responders had EGFR-mutated tumors, with four patients also receiving EGFR tyrosine kinase inhibitor (TKI) therapy alongside PPV. Immune monitoring revealed that five responders developed T-cell responses to EGFR NeoAg peptides due to the vaccine. Moreover, four responding patients had increased frequencies of neoantigen-specific CD8+ T cells in their blood during PPV, indicating T-cell responses were triggered. Combining PPV with EGFR inhibitor therapy was well tolerated, suggesting personalized neoantigen-based vaccination could be a promising immunotherapy for NSCLC patients with EGFR mutations.

### 4.5. Gastrointestinal Cancers Studies

Tran et al. [84] utilized next-generation sequencing and high-throughput immunologic screening to examine tumor-infiltrating lymphocytes (TILs) from 10 patients with metastatic gastrointestinal cancers. Their findings indicated that TILs from 9 out of 10 patients contained CD4+ and/or CD8+ T cells recognizing neo-epitopes derived from mutations in the patient’s own tumor. In total, they identified 17 immunogenic peptide neo-epitopes. This suggests that most patients with these cancers harbor T cells targeting specific tumor mutations, offering promise for tailored vaccines or cell therapies against common epithelial cancers.

### 4.6. Epithelial Cancer Studies

Cafri et al. [85] used a highly sensitive in vitro stimulation and cell enrichment technique to look for memory T cells in the blood of six patients with metastatic cancer. They found memory T cells capable of targeting both unique and shared mutations present in the blood of people with epithelial cancer. While past studies focused on T cells in tumors, this study shows specific T cells exist in the blood, offering a non-invasive way to find and isolate cells or receptors that react to mutations. This discovery opens up new possibilities for using these T cells to develop personalized immunotherapies against epithelial cancers.

Zeng et al. [86] conducted a case study to explore personalized neoantigen-based immunotherapy for treating collecting duct carcinoma (CDC) of the kidney. They focused on an Asian patient with metastatic CDC, which had progressed despite previous Sorafenib treatment. The researchers identified 13 specific neoantigens in the patient’s tumor based on their genetic profile. Using these neoantigens, they developed a personalized treatment plan involving long peptide vaccines and neoantigen-reactive T cells (NRTs). After six cycles of this treatment, the patient showed significant improvements, with stable disease status in tumor burden and relief from bone pain. Tests on blood cells collected after treatment revealed immune responses to 12 of the 13 neoantigens. Biopsy samples taken from CDC sites after three months showed a reduction in mutant allele frequency related to 92% of the neoantigens, suggesting potential elimination of tumor cells carrying these specific neoantigens.

### 4.7. Multiple Tumor Studies

Ott et al. [87] conducted a phase Ib clinical trial to evaluate a personalized neoantigen-based vaccine called NEO-PV-01, combined with the immune checkpoint inhibitor PD-1 blockade (Nivolumab). The trial involved 82 patients with advanced melanoma, non-small cell lung cancer, or bladder cancer. Patients received NEO-PV-01 alongside Nivolumab. After vaccination, all patients showed new neoantigen-specific CD4+ and CD8+ T-cell responses. These T cells were cytotoxic and capable of entering the tumor to kill cells. The study also observed epitope spreading, where T-cell responses expanded to neoantigens not included in the vaccine, potentially boosting the anti-tumor immune response. Importantly, the combination therapy was safe, with no significant treatment-related adverse events reported.

### 4.8. Breast Tumor Studies

Sha et al. [88] aimed to investigate a neoantigen-based vaccine targeting tumor-specific mutated peptides from gene mutations. They studied a 57-year-old woman with a phyllodes tumor who had undergone left breast mass resection. Through high-throughput gene sequencing and detailed analysis, they tailored the vaccine to target the patient’s unique neoantigens. Remarkably, the vaccine led to a complete pathological response in lung metastasis, demonstrating the safety and effectiveness of this immunization approach.

Li et al. [89] developed neoantigen DNA vaccines by combining elongated epitopes with a mutant ubiquitin variant. This fusion aimed to improve antigen processing and presentation. Epitopes were identified using various computational, laboratory-based, and live organism techniques. The study found that the optimized DNA vaccines were immunogenic, generating strong neoantigen-specific immune responses in mice. These immune reactions were comparable to those induced by synthetic long peptide vaccines targeting the same neoantigens. Additionally, when used alongside immune checkpoint blockade therapy, the optimized DNA vaccines showed potential in triggering anti-tumor immunity in preclinical models.

### 4.9. T-Cell Recognition

In their study, Smith et al. [90] showed that common modifications used to improve peptide binding can unexpectedly affect T-cell recognition. They focused on the recognition of both the wild-type (WT) and an anchor-modified version of the well-known gp100209 tumor antigen (ITDQVPFSV) presented by the MHC molecule HLA-A2, using three different gp100209-specific TCRs. Interestingly, they found significant variations in functional responses, highlighting the need for caution when using and interpreting outcomes from anchor-modified peptides. These findings are important because they shed light on the unpredictability of tumor neoantigen immunogenicity, especially in specific scenarios.

The recent increase in studies on peptide-based cancer vaccination highlights the growing interest in this field. Human immunization experiments, which are common in many of these studies, are crucial for assessing the safety and effectiveness of peptide vaccines and evaluating the immune response to potential neoantigens. Most of these studies have emerged in the past few years, showing a clear trend of progress in research. However, human immunization experiments involving neoantigens are still relatively rare. Only a small number of neoantigens have been tested in humans so far, making it challenging to develop accurate prediction tools for immunogenicity. Additionally, many types of cancers have not yet been studied using human immunization experiments. Moreover, many studies involve single-case reports, often with patients in advanced stages of the disease. This makes it difficult to track disease progression or regression after vaccination within a limited timeframe. Despite these challenges, the collective efforts in this field offer a promising future for cancer immunotherapy.

## 5. Databases Containing Human Tumor Antigens

To develop cancer vaccines with optimal safety and efficacy, a thorough investigation of diverse human tumor antigens is imperative. In recent times, remarkable strides have been achieved in this domain, unveiling a plethora of cancer antigens that undergo rigorous validation. This growing repository of validated antigens augments the likelihood of identifying promising candidates for vaccine formulation. Furthermore, the utilization of computational antigen prediction algorithms necessitates access to high-quality experimental data, which is crucial for their effective training. Herein, we delineate some of the prominent human tumor antigen databases currently available, pivotal in facilitating this critical research endeavor.

The Immune Epitope Database (IEDB, https://www.iedb.org/) [94] is a freely available resource funded by NIAID. Although not specific to cancer, the IEDB contains a vast collection of experimentally validated epitopes from various sources, including tumor-associated antigens. As a companion site to the IEDB, The Cancer Epitope Database and Analysis Resource (CEDAR, https://cedar.iedb.org/) [95] provides a freely accessible, comprehensive collection of cancer epitope and receptor data curated from the literature. The Cancer Antigenic Peptide Database (CAPDb, https://caped.icp.ucl.ac.be/) [96] is a comprehensive database of experimentally validated tumor antigens, which includes information on the antigenic peptides presented on major histocompatibility complex (MHC) molecules. The Tumor-Specific Neoantigen Database (TSNAdb v1.0, http://biopharm.zju.edu.cn/tsnadb) [97] is based on pan-cancer immunogenomic analyses of somatic mutation data and human leukocyte antigen (HLA) allele information for 16 tumor types with 7748 tumor samples from The Cancer Genome Atlas (TCGA) and The Cancer Immunome Atlas (TCIA). NEPdb (http://nep.whu.edu.cn/) [98] is a database of T-cell experimentally validated neoantigens and pan-cancer predicted neo-epitopes for cancer immunotherapy. It contains more than 17,000 validated human immunogenic neoantigens and ineffective neo-epitopes within HLAs via curating the published literature with their semi-automatic pipeline. Furthermore, NEPdb also provides pan-cancer level predicted HLA-I neo-epitopes derived from 16,745 shared cancer somatic mutations, using state-of-the-art predictors. The Catalogue of Somatic Mutations in Cancer (COSMIC, https://cancer.sanger.ac.uk/cosmic) [99] is the world’s largest and most comprehensive resource for exploring the impact of somatic mutations in human cancer. It also provides information on tumor antigens resulting from these mutations. A comprehensive online database (TANTIGEN 2.0, http://projects.met-hilab.org/tadb) [100] catalogs 4,296 antigen variants from 403 unique tumor antigens and more than 1500 T-cell epitopes and HLA ligands. It also contains validated TCR sequences specific for cognate T-cell epitopes and tumor antigen gene/mRNA/protein expression information in major human cancers extracted by the Human Pathology Atlas. The cancer antigen database (CAD, http://cad.bio-it.cn/) [101] is a collection of verified cancer antigen peptides crucial to the development of neoantigen-based cancer vaccines. The role of each dataset for algorithm improvement in cancer antigen research, especially neoantigen prediction, is also discussed. A database of T-cell-defined human tumor antigens was developed by Vigneron et al. (www.cancerimmunity.org/peptide) [102]. It compiles all human antigenic peptides described in the literature that fulfill a set of strict criteria needed to ascertain their actual “tumor antigen” nature.

## 6. Summary and Future Perspective

The accumulation of knowledge over the past 15 years concerning tumor immunogenicity suggests a promising future for cancer immunotherapy overall. This progress is evident in the development of monoclonal antibodies, immune checkpoint inhibitors, and cancer vaccines. While the current limitation revolves around the scarcity of in vivo validated data concerning tumor immunogenicity, strides are being made as more human immunization experiments with peptide antigens are conducted. Experimentally validated immunogens will play a crucial role in the development of more robust in silico prediction algorithms, thereby facilitating cancer neoantigen discovery—a process that sets off a positive feedback loop. Ultimately, these advancements hold the potential to pave the way for the first FDA-approved therapeutic peptide cancer vaccine in the near future.

Peptide-based neoantigen cancer vaccines represent a promising frontier in the realm of cancer immunotherapy, owing to their personalized nature and capacity to elicit robust and enduring anti-tumor responses. These vaccines are generally acknowledged for their safety and tolerability in clinical settings. Encouraging outcomes have emerged from clinical investigations, revealing extended overall survival and disease control among patients receiving personalized neoantigen vaccines. Their compatibility with other therapeutic modalities, such as immune checkpoint inhibitors or adoptive T-cell therapies, further bolsters the overall anti-tumor response and augments treatment efficacy. Neoantigen-based vaccines have been probed across various cancer types, rendering them versatile with potential applicability in diverse malignancies. Nevertheless, inherent limitations persist. Tumor antigen heterogeneity and challenges in identification and targeting, as well as limited coverage, diminish the vaccines’ overall efficacy. Ongoing challenges in optimizing vaccine formulations, encompassing adjuvant selection and delivery techniques, may impact their potency. Immune evasion and suppression mechanisms further compromise therapeutic responses. Moreover, the personalized nature of these vaccines incurs higher costs and potential restricted access for certain patients, limiting their widespread adoption. Although initial clinical trials have shown promise, their long-term efficacy and broader benefits in larger patient cohorts warrant further evaluation. Despite these constraints, continual research and progress in cancer immunotherapy strive to address these hurdles, upholding peptide-based neoantigen cancer vaccines as a promising avenue for personalized and targeted cancer treatment. As investigations progress, these vaccines hold substantial potential in enhancing cancer treatment outcomes and patient survival. 

## Figures and Tables

**Figure 1 ijms-25-04934-f001:**
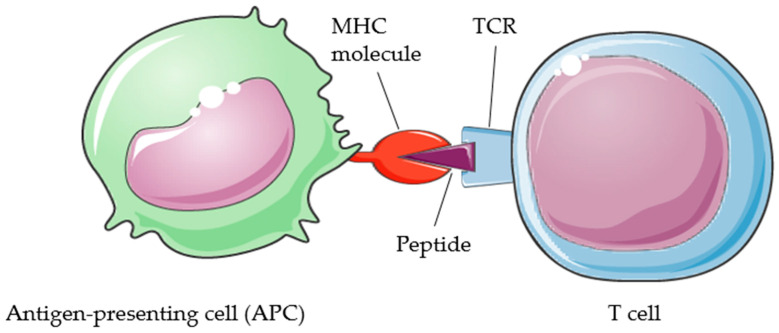
Illustration of T-cell recognition. APC presents the antigen to the T cell in a complex with an MHC molecule. TCR of the T cell recognizes the complex. Other co-stimulatory signals also take part in the process. This figure was produced with the assistance of Servier Medical Art (https://smart.servier.com).

**Figure 2 ijms-25-04934-f002:**
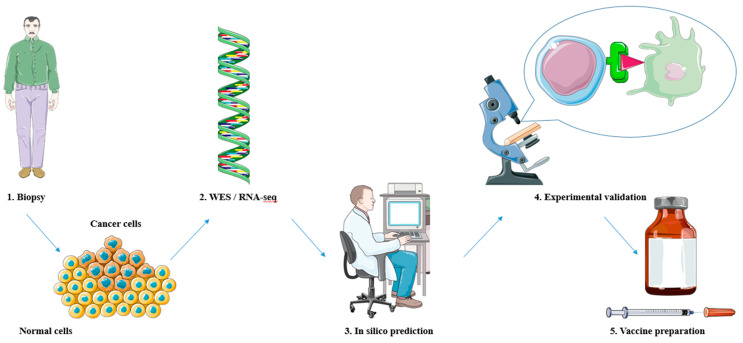
Neoantigen discovery and selection for peptide-based cancer vaccines. 1. Obtaining tumor tissue via biopsy. 2. Whole-exome sequencing (WES) or RNA sequencing (RNA-seq) on tumor DNA or RNA to identify mutations. 3. Neoantigen prediction and prioritization through in silico methods. 4. Experimental validation of predicted neoantigens. 5. The most promising neoantigens are included in vaccine preparations and carried on to preclinical and clinical trials. The figure was produced with the assistance of Servier Medical Art (https://smart.servier.com).

**Table 1 ijms-25-04934-t001:** FDA-approved cancer vaccines.

Vaccine	Trade Names Examples	Target Antigen/s	Cancer Type	Use	Approved Since
Hepatitis B vaccine	Engerix-B^®^, Recombivax HB^®^, Heplisav-B^®^	Hepatitis B virus (HBV) purified surface antigen (HBsAg)	HBV-related hepatocellular carcinoma	Preventive	In use since 1981
Bacillus Calmette–Guérin (BCG) vaccine	Tubervac^®^	Non-pathogenic mycobacterium bovis	Non-muscle invasive bladder cancer (NMIBC), also used as an immune stimulant	Therapeutic	In use since 1990
HPV vaccines	Gardasil^®^, Cervarix^®^, Gardasil 9^®^	Types of L1 protein of human papilloma virus (HPV): Gardasil®—types 6, 11, 16, and 18; Cervarix®—types 16 and 18; Gardasil 9®—types 6, 11, 16, 18, 31, 33, 45, 52, and 58	HPV-associated cervical, oropharyngeal, anal, penile, and vulvovaginal cancers	Preventive	Gardasil^®^—8 June 2006; Cervarix^®^—16 October 2009; Gardasil 9^®^—10 December 2014
Sipuleucel-T	Provenge^®^	Prostatic acid phosphatase	Castration-resistant prostatic cancer	Therapeutic	29 April 2010
Talimogene laherparepvec	T-VEC®, Imlygic^®^	Does not target any antigen/s, directly destroys the cancer cells it infects	Metastatic melanoma	Therapeutic	27 October 2015

**Table 2 ijms-25-04934-t002:** Characteristics of tumor antigens. WT1, Wilms tumor 1; p53, tumor protein 53; PSA, prostate-specific antigen; gp100, glycoprotein 100; NY-ESO-1, New York esophageal squamous cell carcinoma 1; MAGE-A3, melanoma antigen A3; HPV-E6/E7, human papillomavirus; HTLV-1, human T-cell lymphotropic virus type 1.

Type of Tumor Antigen	Description	Tumor Specificity	Central Tolerance	Prevalence in Multiple Patients	Examples
Overexpressed antigens	Antigens found at increased levels in tumors compared with normal healthy cells	Variable	High	High	WT1, p53
Differentiation antigens	Antigens selectively expressed by the cell lineage from which malignant cells evolved	Variable	High	High	PSA, gp100
Cancer/testis antigens	Antigens only been found to be expressed in immune-privileged tissues (testes, fetal ovaries, trophoblast)	High	Low	High	NY-ESO-1, MAGE-A3
Neoantigens	Antigens that are uniquely expressed by cancer cells but not found in normal cells	Complete	None	Variable	Various mutated peptides from different proteins
Viral antigens	Antigens derived from viral proteins that are only present in tumor cells	Complete	None	High	HPV-E6/E7, HTLV-1

**Table 3 ijms-25-04934-t003:** Summary of existing computational methods for tumor neoantigen prediction.

Name	Year	Method ^a^	Availability online (as of 1 March 2024)
**LENS [56]**	2023	Over two dozen separate tools to generate tumor antigen predictions	https://gitlab.com/landscape-of-effective-neoantigens-software
**OpenVax [57]**	2020	Bioinformatics pipeline	https://github.com/openvax
**pVACtools [58]**	2020	Various MHC-I prediction algorithms	https://github.com/griffithlab/pVACtools
**nextNEOpi [59]**	2022	WES/WGS/RNA-Seq pipeline	https://github.com/icbi-lab/nextNEOpi
**TTAgP [60]**	2019	RF	https://github.com/bio-coding/TTAgP
**TIminer [61]**	2017	NGS pipeline	https://bio.tools/timiner
**iTTCA-Hybrid [62]**	2020	RF, SVM	http://camt.pythonanywhere.com/iTTCA-Hybrid
**iTTCA-RF [63]**	2021	RF	http://112.124.26.17:7002/
**TAP [64]**	2021	ML	No
**PSRTTCA [65]**	2023	SCM	http://pmlabstack.pythonanywhere.com/PSRTTCA
**StackTTCA [66]**	2023	Stacking ensemble-learning algorithm	No
**VaxiJen v2.0 [67]**	2007	PSL-DA	http://www.ddg-pharmfac.net/vaxijen/VaxiJen/VaxiJen.html

^a^ WES, whole-exome sequencing; WGS, whole-genome sequencing; RNA-seq, RNA sequencing; NGS, next-generation sequencing; RF, random forest; SVM, support vector machine; ML, machine learning; SCM, scoring card method; PSL-DA, partial least squares discriminant analysis.

**Table 4 ijms-25-04934-t004:** Peptide-based cancer vaccines currently in active, not recruiting clinical trials (participants are receiving an intervention or being examined, but new participants are not currently being recruited or enrolled). Variants of HPV and HBV vaccines are not included.

Vaccine	Description	Cancer Type	Trial Id
FMPV-1	Peptide-based cancer vaccine targeting transforming growth factor beta receptor 2 (TGFBR2)	Colorectal cancer	NCT05238558
TENDU	Synthetic therapeutic peptide conjugate vaccine	Prostate cancer	NCT04701021
UV1	Consists of three long synthetic peptides, representing 60 amino acids of the reverse transcriptase subunit of human telomerase (hTERT)	Malignant pleural mesothelioma	NCT04574583
RO7198457	mRNA-based individualized, therapeutic cancer vaccine targeting an unspecified number of tumor-associated antigens (TAAs)	Pancreatic cancer	NCT04161755
EO2401	Peptide therapeutic vaccine based on the homologies between tumor associated antigens and microbiome-derived peptides	Glioblastoma	NCT04116658
RV001V	Vaccine composed of an immunogenic peptide derived from the Ras homolog family member C (RhoC; Rho-related GTP-binding protein RhoC)	Prostate cancer	NCT04114825
Galinpepimut-S (GPS)	Peptide cancer vaccine comprised of four peptide chains derived from the Wilms’ tumor gene 1 (WT1) protein	Mesothelioma	NCT04040231
AE37	HER2-directed vaccine based on the AE36 hybrid peptide (aa776-790)	Triple-negative breast cancer	NCT04024800
Neoantigen peptide vaccine	Neoantigen peptide vaccines will incorporate prioritized neoantigens and personalized mesothelin epitopes	Pancreatic cancer	NCT03956056
GRT-C903 and GRT-R904	Neoantigen-based therapeutic cancer vaccines based on tumor-specific shared neoantigens, which are immunogenic and unique across a subset of patients	Advanced solid tumors	NCT03953235
UCPVax	Therapeutic cancer vaccine composed of two separate peptides derived from telomerase (hTERT, human telomerase reverse transcriptase)	HPV-positive cancers	NCT03946358
iNeo-Vac-P01	Personalized neoantigen peptide cancer vaccine (5~20 peptides with the length ranging from 15 to 35 amino acids)	Advanced malignant solid tumors	NCT03662815
PVX-410	Multipeptide therapeutic cancer	Breast cancers	NCT03362060
DPX-Survivac	Vaccine composed of survivin-based synthetic peptide antigens and an adjuvant	Lymphomas	NCT03349450
MUC1 peptide-Poly-ICLC	Vaccine composed of 100-amino acid synthetic MUC1 peptide	Lung carcinoma	NCT03300817
H3.3.K27M	Synthetic peptide vaccine specific for the H3.3.K27M epitope	Gliomas	NCT02960230
IMA950	Multipeptide vaccine containing 11 glioma-associated antigens among which 9 are HLA-A*0201-restricted peptides, and 2 are HLA class II-binding peptides	Grade II low-grade glioma (LGG)	NCT02924038
IMU-131	B-cell peptide vaccine composed of a fusion of 3 epitopes from the extracellular domain of HER2/neu conjugated to CRM197 with the adjuvant Montanide	HER2-positive advanced gastric cancer	NCT02795988
Nelipepimut-S (E75)	Nine amino acid sequence from the extracellular domain of the HER2 receptor (residues 369–377 of HER2neu: KIFGSLAFL)	Breast ductal carcinoma	NCT02636582
SurVaxM (SVN53-67/M57-KLH)	Synthetic long peptide mimic peptide that spans amino acids 53 through 67 of the human survivin protein sequence	Glioblastoma	NCT02455557
CMVpp65-A*0201	Antigenic peptide NLVPMVATV	Hematologic malignancies	NCT02396134
HLA-A2-restricted synthetic glioma antigen peptides vaccine	Vaccine consisting of HLA-A2-restricted peptides derived from glioma-associated antigens (GAAs)	Pediatric gliomas	NCT01130077
Bivalent vaccine	Vaccine consisting of two cell-surface antigens (GD2L and GD3L)	Neuroblastoma	NCT00911560

**Table 5 ijms-25-04934-t005:** Studies regarding advancements made in the research field of cancer peptide vaccination.

Subject	Target	Number of Patients	Outcome	Reference
Epitope vaccination	Melanoma	37	Identification of T-cell immunogens in metastatic melanoma. Endogenous responses directed at other melanoma antigens.	Khong et al. [69]
Melanoma antigens (MAGE) vaccination	Melanoma	1	Identification of T-cell immunogens in metastatic melanoma. Antigen spreading.	Corbiere et al. [70]
Dendritic cells vaccination	Melanoma	3	Identification of T-cell immunogens in melanoma. Discovery of previously undetected HLA class I-restricted neoantigens.	Carreno et al. [71]
Identification of T-cell immunogens	Melanoma	8	A methodology to facilitate the isolation of neoantigen-specific T cells derived from tumor and peripheral lymphocytes. Isolation of neoantigen-specific T cells.	Cohen et al. [72]
Mature dendritic cell (mDC) vaccination	Melanoma	4	Identification of T-cell immunogens in melanoma. CD8+ T-cell responses, encompassing multiple neoantigen-specific TCR clonotypes.	Linette et al. [73]
Personalized neoantigen vaccines	Melanoma	8	Identification of T-cell immunogens in melanoma. A long-term persistence of neoantigen-specific T-cell responses following vaccination.	Hu et al. [74]
Personalized RNA-based vaccination	Melanoma	13	Identification of individual mutations, computational prediction of neo-epitopes, and design and manufacturing of a personalized vaccine.	Sahin et al. [75]
KRAS-mutated peptide vaccination	Pancreatic adenocarcinoma	24	Limited immunogenicity.	Abou-Alfa et al. [76]
KRAS-mutated peptide vaccination	Pancreatic adenocarcinoma	1	Identification of immunogenic mutations in KRAS peptides and specific KRAS-targeting TCRs.	Dillard et al. [77]
Personalized multipeptide vaccination	Pancreatic ductal carcinoma	1	Identification of T-cell immunogens. Vaccine induced a multifaceted and persistent immune response.	Sonntag et al. [78]
Neoantigen-derived personalized cancer vaccination	Advanced pancreatic cancer	7	Identification of T-cell immunogens. An extended mean overall survival accompanied by an increase in antigen-specific TCR clones post-vaccination.	Chen et al. [79]
Multi-epitope, personalized neoantigen vaccination	Glioblastoma	10	Identification of T-cell immunogens. Tumor recurrence and the progression of the disease after vaccination.	Keskin et al. [80]
Autologous tumor lysate-dendritic cell vaccine followed by a neoantigen-based synthetic long peptide vaccine	Glioblastoma	1	Identification of T-cell immunogens. Presence of discernible CD8+ and CD4+ T-cell responses specifically directed towards neoantigens induced by the peptide vaccine.	Johanns et al. [81]
Multi-epitope peptide vaccination	Lung squamous cell carcinoma	1	Identification of T-cell immunogens. A substantial reduction in tumor size and positive clinical outcomes.	Li et al. [82]
Personalized neoantigen peptide vaccination	Non-small cell lung cancer with epidermal growth factor receptor (EGFR) mutations	24	Identification of T-cell immunogens. Vaccine-induced T-cell responses directed towards EGFR NeoAg peptides.	Li et al. [83]
Identification of T-cell immunogens	Gastrointestinal cancer	10	Identification of T-cell immunogens. Presence of tumor-mutation-specific T cells.	Tran et al. [84]
Identification of T-cell immunogens	Metastatic epithelial cancer	6	Identification of T-cell immunogens in epithelial cancer. Detected existence of specific T cells in the peripheral blood.	Cafri et al. [85]
Personalized neoantigen-based vaccination and T-cell immunotherapy	Duct carcinoma	1	Identification of T-cell immunogens in advanced collecting duct carcinoma. A reduction in mutant allele frequency corresponding to 92% of the neoantigens.	Zeng et al. [86]
Personalized neoantigen-based vaccine, in combination with Nivolumab	Advanced melanoma, non-small cell lung cancer, bladder cancer	82	Identification of T-cell immunogens in various cancers. Antigen spreading.	Ott et al. [87]
Personalized neoantigen-based vaccine	Breast tumor	1	Identification of T-cell immunogens in phyllodes tumor. Pathological complete response in the lung metastasis.	Sha et al. [88]
Neoantigen vaccination in preclinical models	Murine breast tumor	Mice	Anti-tumor immune responses in preclinical models and neoantigen-specific responses in clinical translation	Li et al. [89]
Effects of peptide anchor modification on T-cell recognition	T-cell leukemia	-	Improved T-cell recognition	Smith et al. [90]

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
