# Peer review of "Tumor-Derived Antigenic Peptides as Potential Cancer Vaccines"

_ijms, 2024, doi:10.3390/ijms25094934_

Round 1
Reviewer 1 Report
Comments and Suggestions for Authors
Authors needs to revise the manuscript.

Reviewer 2 Report
Comments and Suggestions for Authors
The author's manuscript, "The Potential of Tumor-derived Antigenic Peptides in Cancer Vaccines," provides a thorough overview of the state of cancer vaccine research and highlights the integration of immunotherapy strategies with tumor antigens.
Major comments,
The section "Studies with Tumor Antigens" offers valuable insights into the efficacy and safety of tumor-derived antigenic peptides. However, further clarification is needed in several areas to enhance the paper's contribution.
1. Significance of Research Findings: Various studies are outlined, but the broader significance of these findings in cancer vaccine therapy is underdeveloped. Detailing how each study advances our understanding or application of cancer vaccines will provide a clearer overview for readers.
2. Clarification of Concepts: The mention of epitope spreading and endogenous responses requires further explanation. These concepts are crucial for understanding the immune response to cancer vaccines, but their descriptions are brief and somewhat unclear. Expanding on these terms and their relevance to the discussed studies would significantly aid reader comprehension.
3. Comparisons and Contrasts Among Studies: While different research outcomes are briefly mentioned, there is no comprehensive analysis comparing and contrasting these findings. A more detailed discussion of the agreements and discrepancies between studies would offer a richer understanding.
Minor comments
The corrections to be made in the manuscript are as follows:
1. Correction on Page 2, Line 48:
Original: "... while MCH-II molecules..."
Correction: "... while MHC-II molecules...".
2. Correction on Page 3, Line 95:
Original: "...normal localization of CGAs lack HLA class I molecules..."
Correction: "...normal localization of CTAs lack HLA class I molecules...".
These adjustments ensure accuracy in the abbreviations and terminology used, explicitly changing "MCH" to "MHC" (Major Histocompatibility Complex) and correcting "CGAs" (incorrectly used) to "CTAs" (Cancer-Testis Antigens), thus aligning with standard immunological terms and providing clarity in the manuscript.
The reviewer is honored to review the work and believes that addressing these points will strengthen the manuscript's impact. He looks forward to your revised submission.
Comments on the Quality of English LanguageBe careful in stating and using abbreviations.
Reviewer 3 Report
Comments and Suggestions for Authors
The manuscript reviews the recent studies in antigenic peptides as cancer vaccines. Overall, the manuscript is well written and clearly organized. There are two concerns that should be addressed before publication. First, the computational methods to predict and prioritize antigens are scantly mentioned at page 5. The reviewer thinks that these methods should be more extensively described and maybe illustrated by one Figure. Second, section 4 should be reorganized because the reviewer thinks that a review cannot be a mere list of papers in which short abstracts for each study are simply reported. A review should tell a story in which the reviewed papers are arranged and critically evaluated (if necessary) to describe how the research evolved during the years and which are the possible future developments in the field. Hence, the reviewer suggests to extend the Conclusions to better describe the future perspectives.
Reviewer 4 Report
Comments and Suggestions for Authors
March 10, 2024
Manuscript ID: ijms-2891907
Type of manuscript: Review
Title: Tumor-derived antigenic peptides as potential cancer vaccines
Authors: Stanislav Kostadinov Sotirov, Ivan Dimitrov Dimitrov
Overview and general recommendation:
This paper provides a good summary of the status of the research on tumor-derived antigenic peptides as cancer vaccines. This will help both expert and non-specialist readers to follow up on relevant research. The paper is well written, and data provided and summarized nicely, and thus can be published as is.
I have only very miner comments.
(1) The authors would benefit from adding a schematic illustration of p-MHC and TCR interactions at the begging of the paper.
(2) “Summary and future perspective” might be better than “Conclusion”.
Round 2
Reviewer 3 Report
Comments and Suggestions for Authors
The Authors revised the manuscript addressing the concerns raised by the initial submission and now the paper can deserve publication
Author Response
We would like to thank the reviewer for reviewing our manuscript.